# A polycistronic system for multiplexed and precalibrated expression of multigene pathways in fungi

Qun Yue[1,8], Jie Meng[2,8], Yue Qiu[2], Miaomiao Yin[1], Liwen Zhang [1], Weiping Zhou[3], Zhiqiang An [4], Zihe Liu [2], Qipeng Yuan [2], Wentao Sun [5], Chun Li [5], Huimin Zhao [6], István Molnár [7,9] ✉, Yuquan Xu [1,9] ✉ & Shuobo Shi [2,9] ✉

Synthetic biology requires efficient systems that support the well-coordinated co-expression of multiple genes. Here, we discover a 9-bp nucleotide sequence that enables efficient polycistronic gene expression in yeasts and filamentous fungi. Coupling polycistronic expression to multiplexed, markerless, CRISPR/Cas9-based genome editing, we develop a strategy termed HACKing (Highly efficient and Accessible system by CracKing genes into the genome) for the assembly of multigene pathways. HACKing allows the expression level of each enzyme to be precalibrated by linking their translation to those of host proteins with predetermined abundances under the desired fermentation conditions. We validate HACKing by rapidly constructing highly efficient *Saccharomyces cerevisiae* cell factories that express 13 biosynthetic genes, and produce model endogenous ($1{,}090.41 \pm 80.92$ mg L$^{-1}$ squalene) or heterologous ($1.04 \pm 0.02$ mg L$^{-1}$ mogrol) terpenoid products. Thus, HACKing addresses the need of synthetic biology for predictability, simplicity, scalability, and speed upon fungal pathway engineering for valuable metabolites.

Synthetic biology demands systems that support the efficient and well-tuned co-expression of multiple genes[1,2]. As opposed to prokaryotes, the overwhelming majority of genes in eukaryotes are transcribed to monocistronic mRNA that are then translated into single proteins[3]. Designing efficient and widely applicable polycistronic expression for eukaryotes would reduce the complexity of genetic constructs, simplify multistep strain engineering, and facilitate synchronized expression of pathway enzymes at predetermined levels under the desired fermentation conditions[1,4]. Current strategies for polycistronic expression in eukaryotes take advantage of internal ribosomal entry sites (IRESs), or self-cleaving 2 A peptides[4]. IRESs recruit ribosomes to

start cap-independent translation of a second open reading frame (ORF) at an internal initiation site within a polycistronic mRNA[5,6]. However, the low efficiency of IRESs has prevented their widespread application in metabolic engineering and synthetic biology[4]. 2 A peptides direct the production of independent proteins from a single mRNA using a ribosomal skipping mechanism[4,7]. The utility of 2 A peptides has been demonstrated in several metabolic engineering applications, such as the production of *C*-glucosylflavones[8] or *β*-carotene[9]. The ribosomal skipping efficiency of 2 A peptides is up to 80%[10,11], and the addition of a GSG motif in front of the 2 A peptide sequence increases this efficiency close to 100%[12]. However, even the

[1]Biotechnology Research Institute, The Chinese Academy of Agricultural Sciences, Beijing, China. [2]Beijing Advanced Innovation Center for Soft Matter Science and Engineering, College of Life Science and Technology, Beijing University of Chemical Technology, Beijing, China. [3]University of Chinese Academy of Sciences, Beijing, China. [4]Texas Therapeutics Institute, the Brown Foundation Institute of Molecular Medicine, University of Texas Health Science Center at Houston, Houston, USA. [5]Key Lab for Industrial Biocatalysis, Ministry of Education, Department of Chemical Engineering, Tsinghua University, Beijing, China. [6]Department of Chemical and Biomolecular Engineering, University of Illinois at Urbana-Champaign, Urbana, USA. [7]VTT Technical Research Centre of Finland, Espoo, Finland. [8]These authors contributed equally: Qun Yue, Jie Meng. [9]These authors jointly supervised this work: István Molnár, Yuquan Xu and Shuobo Shi. ✉e-mail: istvan.molnar@vtt.fi; xuyuquan@caas.cn; shishuobo@mail.buct.edu.cn

fully processed proteins will have extra peptides appended to their termini that could affect their structures and functions[11]. Thus, current solutions for polycistronic expression do not satisfy the need for high efficiency, fidelity and throughput necessary for the facile and predictable expression of complex biosynthetic pathways, such as those of natural products.

In this work, we demonstrate an interesting way to build polycistronic sequences in yeasts and filamentous fungi. We identify a 9-bp nucleotide sequence that enables the efficient translation of more than one protein from a polycistronic mRNA. When combined with multiplexed CRISPR-based genome editing, this sequence allows the construction of multiple synthetic bicistrons where the translation of genes of interest (GOI) are coupled to those of endogenous genes of the host. Since the partner open reading frames in the bicistron are co-transcribed under the control of the same native promoter, the expression of the upstream, native gene (the driver) determines that of the downstream GOI in a predictable manner. This way, omics data gathered under the desired cultivation conditions from the host may serve as a guide to identify appropriate driver genes for the controlled expression of the GOI. We validate this system, termed HACKing (a Highly efficient and Accessible system by CracKing genes into genome), by rapidly attaining the high-level production of squalene and mogrol in the yeast *Saccharomyces cerevisiae*. Thus, this work demonstrates a facile synthetic biology strategy for the assembly and controlled expression of multigene metabolic pathways in industrial chassis organisms.

## Results

### A fungal intergenic sequence mediates bicistronic transcription

With the exponential accumulation of genome sequences in databases, an increasing number of polycistronic gene transcripts are discovered in eukaryotic species[13,14], with different mechanisms. Recently, we reported a short intergenic sequence (*IGG1*) that connects separate genes into a functional operon in the filamentous fungus *Glarea lozoyensis*[15]. To test whether *IGG1* could also mediate the expression of different coding sequences (CDSs) transcribed to bicistronic messages in *S. cerevisiae*, we used *IGG1* to link the *GFP* reporter to the genomic copy of the glycolytic gene *TDH3* that contained a C-terminal stop codon (Fig. 1a,b). A GFP signal was detected in *TDH3::IGG1-GFP* transformants but not in the parent strain without *GFP* or in the *TDH3::GFP* transformants without the *IGG1* sequence (Fig. 1a and Supplementary Fig. 1). The growth of the *TDH3::IGG1-GFP* transformants remained unchanged (Supplementary Fig. 2), suggesting continued expression of *TDH3* whose knockout would have led to a significantly decreased growth rate[16]. A cDNA encompassing both the *TDH3* and the *GFP* CDSs was amplified from *TDH3::IGG1-GFP* transformants by reverse transcription (RT)-PCR (Fig. 1b). These results verified the existence of a single, co-transcribed transcript for this bicistronic cassette, and the translation of functional proteins for both constituent messages making up the bicistron.

### Optimization and characterization of IGG-mediated expression

Optimization of *IGG1* resulted in a mere 9-bp sequence (*IGG6*) that afforded the highest GFP fluorescence among the *TDH3::IGG_X-GFP* bicistronic transcripts, representing a 84-fold increase of the signal over that of the original *IGG1*-based construct (Fig. 1a). We validated the formation of a single transcript from the *IGG6*-mediated bicistron by RT-PCR (Fig. 1b). To compare the efficacy of *IGG6* with those of known IRESs or 2 A peptides, six IRESs with high ribosome recruiting activities and two highly efficient 2 A peptides were selected from previous studies to construct bicistrons *TDH3::IRES_X-GFP* or *TDH3::2A_X-GFP*, respectively (Supplementary Table 1)[17,18]. The GFP fluorescence intensity from the *IGG6*-coupled bicistron was 12 to 130-fold higher than those mediated by IRESs, and 37 to 47% of those mediated by 2A peptides (Supplementary Fig. 3).

To confirm that *IGG6* effects the formation of distinct, separate proteins and not fused ones, we designed an *IGG6*-mediated bicistron that is composed of *GFP* and *mCherry* featuring distinct subcellular localization tags. As expected, both fluorescent proteins were detected in the same cells with *mCherry::IGG6-GFP*, but the signals localized to different cellular compartments (Fig. 1c). Expression of a fusion protein created from these two reporters (*mCherry-fusion-GFP*, without a stop codon and with no *IGG* between the two ORFs) led to the co-localization of the two signals, while *mCherry::GFP* transformants without *IGG* but with a stop codon separating the two ORFs produced only the mCherry signal, as expected (Supplementary Fig. 4).

Encouraged by these results, we constructed polycistronic cassettes with *IGG6* in which the kanamycin/Zeocin resistance gene *KanR* was adopted as the second, third or fourth ORF, respectively. *S. cerevisiae* transformants with such constructs were able to grow on Zeocin, validating the successful translation of this selectable marker from the resulting polycistronic transcripts (Supplementary Fig. 5). In addition to Zeocin resistance, the production of β-carotene and phytoene was also confirmed in the engineered yeast K4 that carries a tetrascistron with the carotenoid biosynthetic genes *crtYB*, *crtE*, and *crtI*, together with *KanR* (Supplementary Fig. 6). This validated the expression of the first three ORFs in addition to *KanR*[19]. Next, we tested the functionality of *IGG6* in different fungi. To form the *IGG6*-mediated bicistron, *GFP* was linked by *IGG6* to *TDH3* in yeast *Pichia pastoris*, *TEF1* in yeast *Yarrowia lipolytica*, and phosphopantetheinyl transferase gene *NpgA* in filamentous fungus *Aspergillus nidulans*, respectively. GFP signals were observed in the transformants of these three fungi by fluorescence microscopy (Supplementary Fig. 7). In addition, a band of the expected size for GFP (~27 kDa) was also observed in Western blotting for *P. pastoris* using an anti-GFP antibody (Supplementary Fig. 8). Taken together, these data demonstrate that *IGG6* is a broad-spectrum intergenic signal that enables polycistronic expression in fungi.

### IGG6 mediates translation re-initiation

To clarify the mechanism of *IGG6*, we designed a series of bicistronic constructs with a translation blocking sequence (TBS)[20] incorporated at different positions (Fig. 2a). *TDH3* translation was successfully verified by Western blotting for all constructs except for the one where the TBS preceded the start codon of *TDH3*, as expected (Fig. 2a, Supplementary Fig. 9). However, no GFP signal was evident with any construct with a TBS, even where the TBS sequence was upstream of *TDH3*. This indicates that the downstream message cannot be translated without the translation of the upstream message; and no new ribosome is recruited to translate the downstream message from the bicistron. A GFP signal was duly detected when the *IGG6*-linked bicistron had no TBS. This signal did not originate from an inadvertent Tdh3p-GFP fusion as the *TDH3* gene contained a stop codon downstream of the FLAG peptide in all constructs, and the anti-FLAG antibody detected only a 37 kDa band corresponding to Tdh3p (Supplementary Fig. 9). In contrast, while a Tdh3p-FLAG-GFP fusion protein (with no *IGG* sequence between the fusion partners) produced a GFP signal, it was detected only as a 64 kDa band by the anti-FLAG antibody (Supplementary Fig. 1).

Taken together, we propose that the expression of the distal gene in *IGG6*-mediated bicistronic cassettes results from translation re-initiation (Fig. 2b). In contrast to IRESs, the ribosome may not dissociate from the mRNA after terminating the translation of the upstream message but would rather resume scanning and re-initiate translation at the start codon of the downstream message. Meanwhile, the presence of stop codons at the 3′ end of each ORF ensures that fusion proteins are not generated, and spurious peptide tails are not appended to the proteins during *IGG6*-mediated bicistronic translation.

### Bicistronic expression of genes of interest

Based on these results, we envisaged a system that would allow a Gene of Interest (GOI) to be integrated into the genome of a chassis

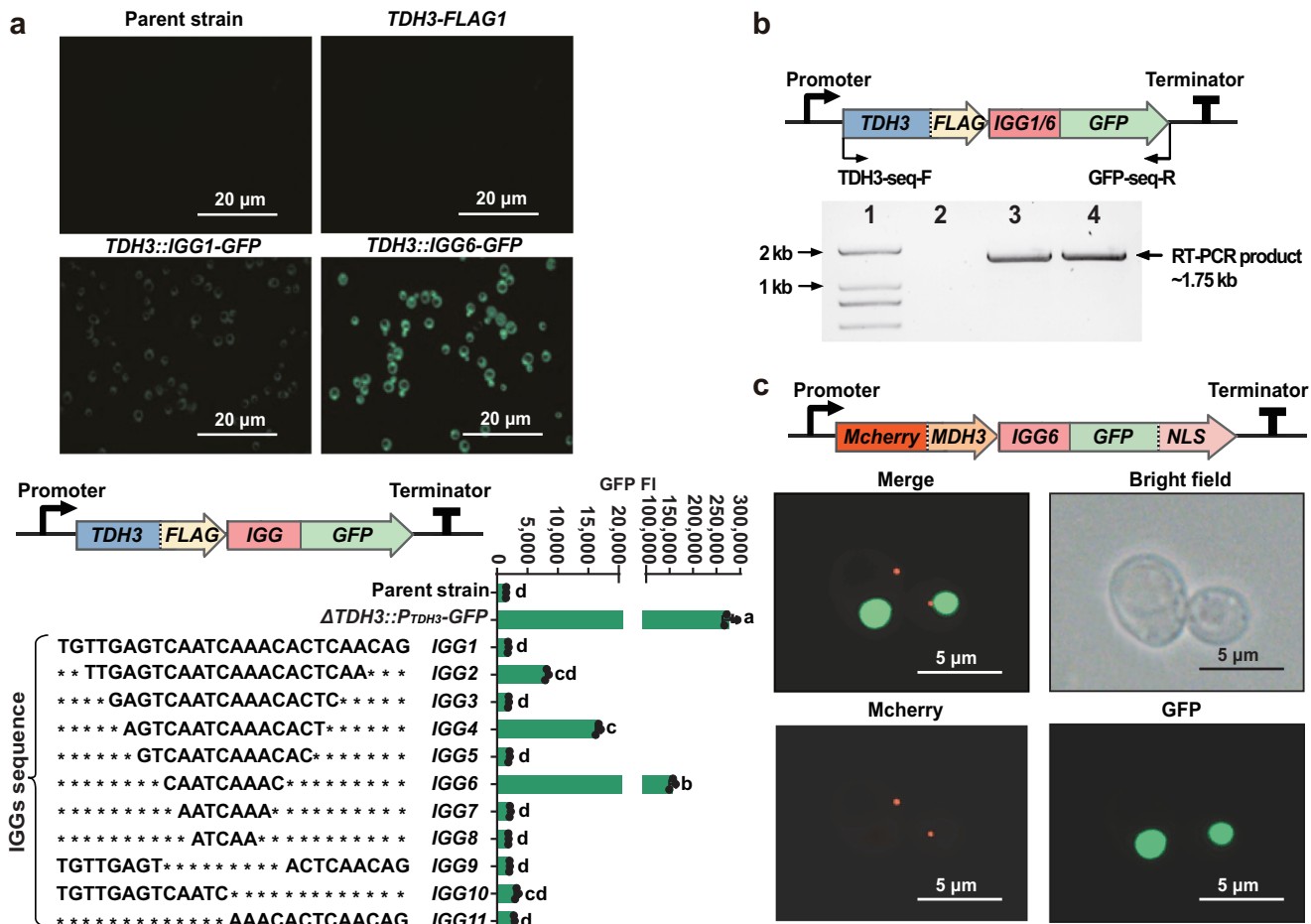

**Fig. 1 | Expression of a distal gene in bicistronic cassettes constructed with intergenic sequence IGGs. a** Verification and optimization of IGG-mediated polycistronic expression. A DNA fragment for FLAG(stop codon)-T$_{ADH2}$-TRP1 was introduced to replace the stop codon and 45 bp of the 3′ untranslated region (UTR) of the genomic copy of the constitutive gene TDH3 by gene knock-in via homologous recombination (TDH3-FLAG1), using S. cerevisiae CEN.PK2−1D as the parent strain. Alternatively, FLAG(stop codon)-IGG$_X$-GFP-T$_{ADH2}$-TRP1 was introduced to the same position by gene knock-in via homologous recombination, with the TDH3 and the GFP genes linked by various IGG sequences (IGG1 - IGG11). Expression of GFP was verified by fluorescence microscopy, as shown for transformants with TDH3::IGG1-GFP and TDH3::IGG6-GFP. The scale bars for the images are 20 μm. The fluorescence intensities of GFP (GFP FI), mediated with different IGGs were quantified, with the parent S. cerevisiae strain serving as the negative control. For a positive control, the TDH3 coding sequence (CDS) was replaced with that of GFP, placing GFP directly under the control of the TDH3 promoter as a monocistron (ΔTDH3::P$_{TDH3}$-GFP). Data and error bars show the mean and standard deviation of three independent biological replicates. Statistical analysis was performed by one-way ANOVA test with Tukey Pairwise Comparisons (95% Confidence). Means that do not share a letter are significantly different ($p < 0.05$). **b** Transcription of bicistronic cassettes TDH3-IGG1-GFP and TDH3-IGG6-GFP detected by RT-PCR with primer set TDH3-seq-F and GFP-seq-R (product size: 1.75 kb). Lane 1, DNA size marker; lane 2, negative control (S. cerevisiae CEN.PK2-1D); lane 3, TDH3::IGG1-GFP transformant; lane 4, TDH3::IGG6-GFP transformant. **c** mCherry with an MDH3 peroxisome localization tag, and GFP with a SV40 nuclear localization tag were encoded to a bicistronic transcript linked by IGG6, and the expression of the two proteins were observed by fluorescence microscopy. The scale bars for the images are 5 μm. Source data are provided as a Source Data file.

organism in the form of a bicistronic transcription unit formed with a selected host gene. Such a bicistronic unit would be expressed by the endogenous promoter of the host gene, and translation of the GOI (the second CDS in the bicistron) would be linked to that of the host CDS by IGG6. This way, host genes with appropriate expression levels under the desired cultivation conditions could be selected to serve as the drivers for the hitched GOI. To validate this concept, we needed to select appropriate drivers for bicistronic GOI expression, and validate the correlation between the expression of the native gene and that of the GOI in the bicistronic units.

To demonstrate a potential workflow for such a selection process, we performed a proteome analysis of the S. cerevisiae chassis under exponential growth in a rich medium during aerobic cultivation (Methods; Supplementary Data 1). We selected 104 candidate driver genes based on their high expression level, and the presence of a protospacer and protospacer adjacent motif (PAM) region near the 3′

ends of the genes (Supplementary Note 1, Supplementary Data 2). Next, we used CRISPR/Cas9 to facilitate the integration of IGG6-linked GFP cassettes downstream of the 3′ ends of these target genes (Fig. 3a, b). We abandoned 39 candidate drivers since their designed gRNA sequences turned out to be invalid (Supplementary Data 3). Using the remaining 65 integration sites, we obtained a library of yeast strains with synthetic GFP bicistrons, all of which yielded easily detectable GFP signals (Fig. 3c, Supplementary Data 4). Evaluation of the GFP fluorescence intensities after ten rounds of subculture revealed no significant changes for any of the strains, verifying the stability of the bicistronic expression system (Fig. 3c). The GFP fluorescence intensities presented a high-positive correlation with the expression intensities of the driver genes measured in the proteomics experiment ($R^2 = 0.7644$, Supplementary Fig. 10). We also quantified the expression of the driver-FLAG and the GFP proteins by ELISA in several driver-FLAG::IGG6-GFP transformants (Supplementary Fig. 11a).

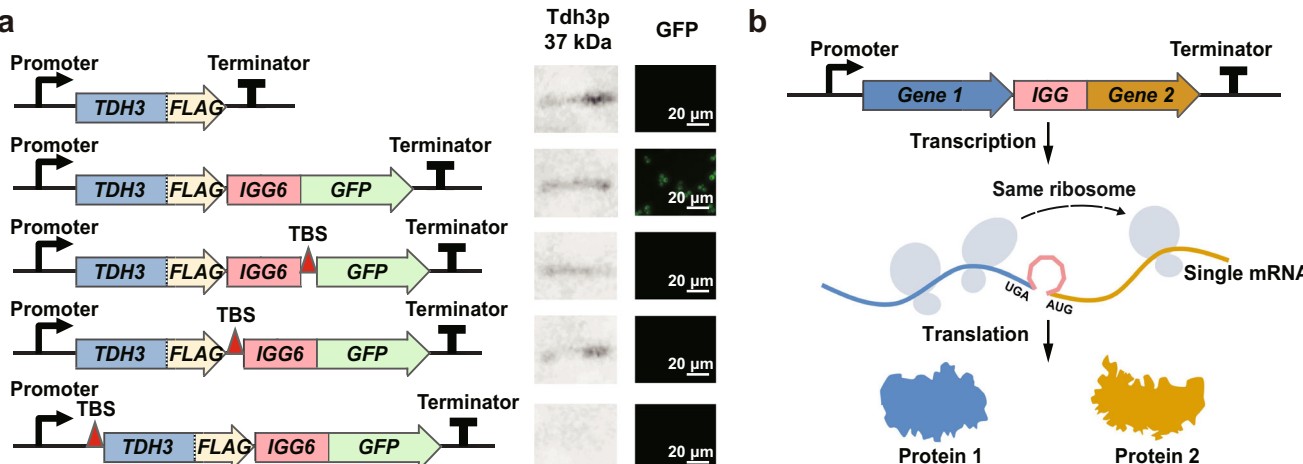

**Fig. 2 | Translation of *IGG*-mediated bicistronic cassettes. a** Translation re-initiation at the distal gene *GFP*, mediated by *IGG6*. Translation of Tdh3p and GFP from different gene expression units was monitored by Western blotting and fluorescence microscopy, respectively. The scale bars for the images are 20 µm. Red triangles represent a translation blocking sequence (TBS). Uncropped Western blots are available (Supplementary Fig. 9). **b** Model for the role of *IGG* during bicistronic expression in eukaryotes. Two contiguous genes linked by *IGG* are co-transcribed into a bicistronic mRNA under the control of the same promoter. The mRNA is then translated into two individual proteins by the ribosome which is not released after translation of the upstream open reading frame (ORF), but rather resumes scanning and initiates the translation of the downstream ORF.

In *TDH3-FLAG::IGG6-GFP*, the accumulation of GFP is about 55% of that of Tdh3p. Similar proportions (31–76% of the driver) were observed in 75% of the selected *driver-FLAG::IGG6-GFP* transformants. To evaluate the effect of *IGG6-GFP* introduction on the expression of the driver protein, several *driver-FLAG* transformants were also constructed using CRISPR/Cas9, and the expression of the driver protein in these transformants and their corresponding *driver-FLAG::IGG6-GFP* bicistronic transformants were quantified by ELISA. The accumulation of the driver proteins in the *driver-FLAG::IGG6-GFP* bicistronic transformants were moderately reduced, reaching 38–64% of that in their corresponding *driver-FLAG* transformants (Supplementary Fig. 11b).

## HACKing allows efficient production of valuable products in yeast

With a library of 65 validated driver genes at hand, we considered the use of GTR-CRISPR[21] for the rapid, multiplexed integration of multiple GOIs into multiple bicistronic transcription units, without the requirement of selection markers (Fig. 4a). Combining multiplexed CRISPR to create synthetic bicistronic transcriptional units with *IGG6*-facilitated co-translation (a system we termed HACKing [Highly efficient and Accessible system by CracKing genes into the genome]) would then allow the simultaneous expression of the GOI-encoded enzymes at stable, precalibrated levels (Fig. 4a; Supplementary Note 1). We validated the HACKing system by demonstrating the facile, rapid generation of a *S. cerevisiae* strain that produces high amounts of the triterpene squalene. Further, we extended the engineered squalene pathway to produce mogrol, a high-value plant triterpenoid (Fig. 4b).

Squalene not only finds various applications in the pharmaceutical and personal care industries, but also serves as a precursor for the biosynthesis of many triterpenoids[22,23]. Eight genes (six from *S. cerevisiae* and two from *Enterococcus faecalis*) that collectively encode a complete mevalonate (MVA) pathway were chosen for overexpression with the HACKing system[24,25]. All the eight pathway enzymes were targeted for peroxisomal localization to avoid further processing by the endogenous squalene monooxygenase Erg1p[24] (Fig. 4b, Supplementary Table 2). Within 2 weeks, two rounds of multiplexed integrations were completed, generating the engineered strain HCS1. HCS1 produced 1,090.41 ± 80.92 mg L$^{-1}$ squalene upon shake-flask cultivation, a yield that is about 900 folds higher than that of the parental strain and is on par with that reported recently[26] (Fig. 4c,

Supplementary Data 5). The drug terbinafine reduces the oxidation of squalene through inhibition of Erg1p, and thus facilitates the higher accumulation of squalene produced by the cytosolic (but not the engineered peroxisomal) MVA pathway[27]. Squalene production in HCS1 did not show a significant change after supplementing the fermentation with terbinafine, in agreement with the targeted peroxisomal localization of the overexpressed MVA pathway in our strain.

Mogrol is an important precursor of the sweetener mogroside V and displays neuroprotective and memory impairment-attenuating activities[28–30]. We used the HACKing system to express a panel of plant genes[31] in the peroxisome to achieve the heterologous production of mogrol in *S. cerevisiae* HCS1 as the chassis (Fig. 4b). First, the mogrol pathway panel of genes *SgCYP87D18*, *SgCDS*, *SgSQE1*, *SgEPH3*, and *AtCPR*[31] were integrated to form bicistronic transcription units with driver genes that showed very high expression, to generate strain HCM1 (Supplementary Table 3). To verify the dependence of mogrol productivity on the strength of the driver genes, we also constructed strains HCM2 and HCM3 where two other sets of driver genes were selected for the *SgCDS*, *SgSQE1*, and *SgEPH3* genes (Supplementary Table 3). These additional driver sets showed medium and low expression, respectively. Within 2 weeks, two rounds of multiplexed genome editing afforded the engineered yeast strains HCM1 to HCM3 that produced 1.04 ± 0.02, 0.54 ± 0.01, and 0 mg L$^{-1}$ mogrol in shake-flask fermentations, respectively (Fig. 4d, Supplementary Data 5). This verified the successful functional expression of the five-gene mogrol biosynthetic module using the HACKing system, with remarkable ease and high efficiency (~100 times higher mogrol production with strain HCM1 than that recently reported for *S. cerevisiae* as the host)[32]. These experiments also validated that production levels can be tuned by the appropriate selection of drivers, relying on the predetermined translation levels. In addition to mogrol, strain HCM1 still produced 884.12 ± 35.88 mg L$^{-1}$ squalene (81% of that of HCS1), implying that further increases in mogrol production may be possible upon additional rounds of strain engineering.

To assess the cellular-level response to bicistronic expression, transcriptome analysis was performed with the parent strain CEN.PK2-1D and the mogrol-producing strain HCM1 (Supplementary Data 6). The genome alignment tracks of RNA-seq data confirmed the existence of bicistronic mRNA for the transgenes linked to the endogenous driver genes by *IGG6* (Supplementary Fig. 12). Among the 13 driver

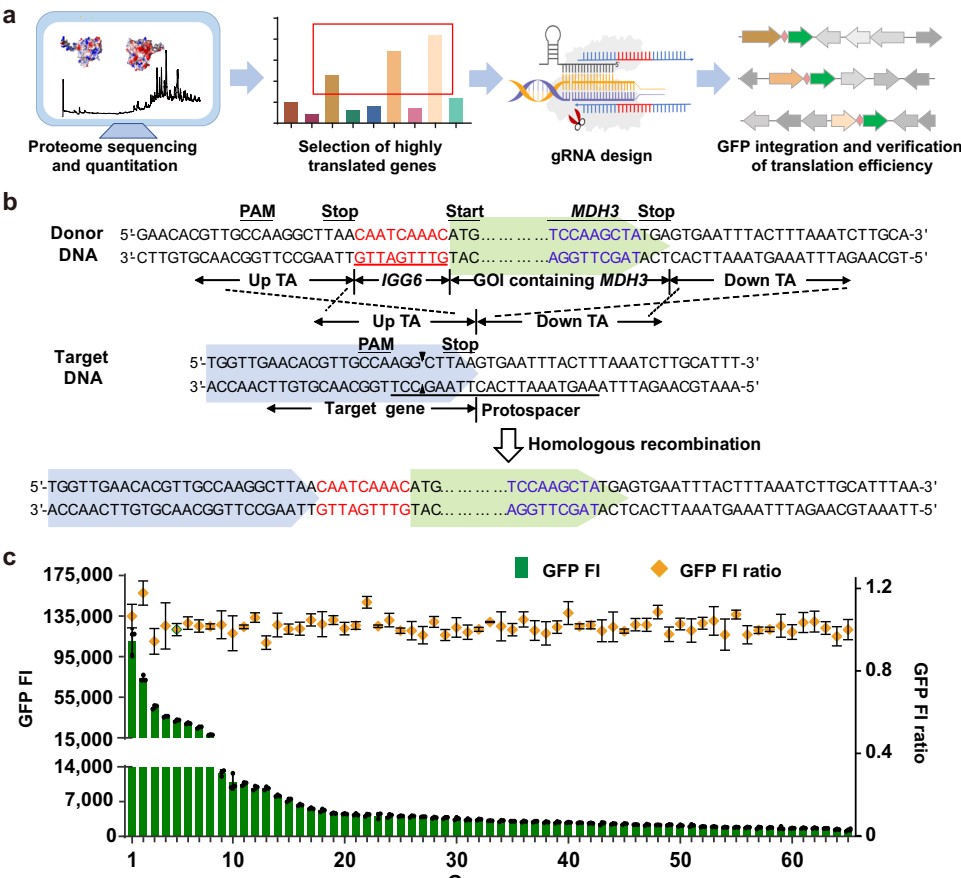

**Fig. 3 | Development and characterization of the bicistronic expression system.**
**a** Schematic workflow to identify host genes (drivers) translated at appropriate, predetermined levels under the desired cultivation conditions, to which genes of interest (GOI, exemplified by GFP in *green*) may be appended (hitched) with *IGG6* using CRISPR/Cas9 to form artificial bicistrons. Arrows indicate the direction of the gene. *Pink rectangles*, the *IGG6* sequence. **b** Design of the integration. Taking the *TDH3* gene as an example of a driver, a gRNA is designed to the 3′ end of the *TDH3* coding sequence (CDS) and a double stranded break is introduced within 5 bp from the stop codon by CRISPR/Cas9. The donor DNA contains four sections: upstream targeting arm (Up TA), *IGG6*, GOI (with an *MDH3* peroximal targeting sequence as an example) and downstream TA (Down TA). The TA, *IGG6* and *MDH3* sequences

are introduced with appropriate synthetic primers during PCR amplification. Homologous recombination places the *IGG6*-linked GOI between the stop codon and the terminator of *TDH3* to form a synthetic bicistronic cassette expressed under the control of the *TDH3* promoter. **c** Fluorescence intensity (FI) of cells carrying a *GFP* hitched to various driver genes with *IGG6* using CRISPR/Cas9. Gene names and their corresponding numbers are listed in Supplementary Data 4. *GFP FI ratio*: Fluorescence intensity of the first-generation strain divided by that of the tenth-generation strain. Data and error bars show the mean and the standard deviation of three independent biological replicates. Source data are provided as a Source Data file.

genes that were used for the bicistronic expression of the terpenoid pathway genes (eight for the squalene and five for the mogrol module), seven driver genes were moderately downregulated in HCM1 (Supplementary Fig. 13). The transcription of the remaining six driver genes showed no significant changes, although a trend for decrease was noticed. These changes may be due to the increased length of the bicistronic transcripts compared to the monocistronic ones for the drivers[33]. However, increased transcript lengths may provide more time for translation, partially offsetting the reduction in mRNA levels[34]. In contrast, the transcription of the genes bracketing the drivers rarely presented significant changes in the mogrol-producing strain HCM1.

## Discussion

Based on the recently discovered *IGG1* sequence of *G. lozoyensis*[15], we developed a 9-bp nucleotide sequence, *IGG6*, to engineer the facile bicistronic expression of genes of interest (GOI) in a variety of fungi. Compared to IRESs, *IGG6* leads to a stronger expression of distal GOI in bicistrons. Although this expression intensity is lower than that mediated by some 2 A peptides, *IGG6* still offers distinct advantages for polycistronic expression of more genes. First, *IGG6*-mediated translation does not append any additional amino acids that could affect the

functions of either protein encoded by the bicistron[11]. Second, *IGG6* is much shorter than 2 A peptides which usually contain 18–22 amino acids[12]. In many cases, various tags, such as those for subcellular localization, need to be appended to the GOI. The DNA sequences encoding 2 A peptides are too long to be added to the primers that also encode long tags, such as those for the mitochondrial localization tag MLS26[35,36]. This then necessitates multistep cloning procedures, instead of a single-step PCR amplification of the GOI for CRISPR/Cas9-mediated integration. Third, the viral origin of 2 A peptides may raise regulatory concerns, not applicable to *IGG6* that originates from the nonpathogenic fungus *Glarea lozoyensis*. With these advantages, we applied the *IGG6* sequence in conjunction with multiplexed CRISPR/Cas9 to enable the markerless in situ construction of multiple bicistronic cassettes in the genome of a model host. The resulting HACKing system was then validated by demonstrating the highly efficient expression of multigene pathways, generating *S. cerevisiae* cell factories that produce large amounts of high value terpenoids such as squalene and mogrol.

The HACKing system does not require the painstaking refactoring of every transgene in a biosynthetic pathway with host-specific upstream activating sequences, promoters, and terminators, nor

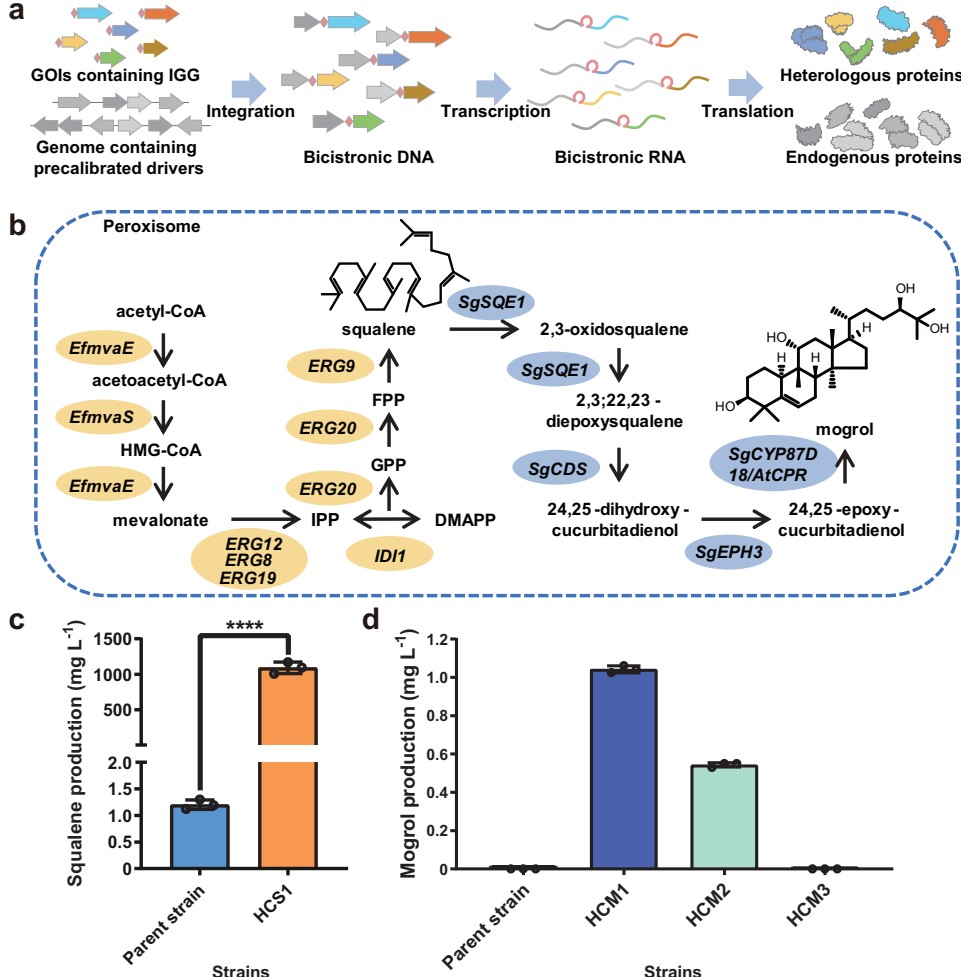

**Fig. 4 | The HACKing system enables efficient construction of cell factories.**
**a** The HACKing system: A host gene with an appropriate, pre-validated translation level under the desired cultivation conditions is selected for each GOI to serve as a driver. Using a PCR amplification reaction, each GOI is appended with an IGG sequence and short targeting arms for integration. The GTR-CRISPR technique[21] is used for multiplexed genome editing that integrates each GOI amplicon downstream of its selected endogenous gene driver to generate multiple bicistrons. This creates strains where multiple GOI are expressed as bicistronic transcripts at precalibrated levels, and translated into proteins with no extraneous sequences to reconstitute biosynthetic pathways. Arrows indicate the direction of the gene. *Pink rectangles*, the *IGG6* sequence. **b** Metabolic pathways to produce squalene (highlighted in *orange*) and mogrol (in *cyan*) in the engineered yeast. Squalene biosynthesis starts from acetyl-CoA and features ten enzymatic reactions (encoded by eight selected genes). Mogrol biosynthesis involves five enzyme-catalyzed reactions, starting from squalene. **c** Squalene production in shake flask fermentations after 6 days of cultivation with the parental *S. cerevisiae* CEN.PK2-1D strain or the HCS1 strain obtained by HACKing. Data and error bars show the mean and the standard deviation of three independent biological replicates. The *p-value* was obtained by two-tailed *t*-test. ****$p < 0.0001$. **d** Mogrol production in shake flask fermentations after 2 days of cultivation with the parental *S. cerevisiae* CEN.PK2–1D strain or the HCM1, HCM2, and HCM3 strains obtained by HACKing. Data and error bars show the mean and the standard deviation of three independent biological replicates. Source data are provided as a Source Data file.

does it involve the assembly of multigene expression cassettes from such refactored gene cassettes. Instead, it requires only PCR amplification of the multiple GOI with appropriate primers, followed by multiplexed CRISPR genomic knock-in. HACKing does not create knockout mutants by replacing native genes on the chromosome of the chassis. It does not require the identification of multiple landing sites for the transgenes in transcriptionally active regions whose disruption is nevertheless physiologically neutral under the intended fermentation conditions. Instead, it allows for the flexible and stable expression of multiple genes at precalibrated levels and predetermined timing by hitching their expression to those of characterized driver genes of the host. Depending on the needs of the researcher, these drivers may be chosen from those host genes that are highly expressed under a broad set of conditions. Conversely, users may select drivers that are expressed only under certain physiological states or environmental/process conditions, or those that are activated

or repressed in the presence of specific elicitors. Since the protein products of the driver genes are presumably important for the host under the selected fermentation conditions, they may not be easily silenced nor dramatically repressed. This then secures the expression of the hitched, downstream GOI, coupled to the driver by *IGG6* within bicistronic or polycistronic transcription units. Thus, HACKing is a promising strategy that is expected to facilitate sophisticated applications in synthetic biology by improving our capacity for facile and potentially automatable multigene pathway prototyping in a variety of fungal chassis designed for industrially relevant fermentation process conditions.

## Methods
### Strains, plasmids, and culture conditions
*Saccharomyces cerevisiae* CEN.PK2-1D, *Pichia pastoris* GS115, *Yarrowia lipolytica* PO1f and *Aspergillus nidulans* A1145 ΔEMΔST were used as

the hosts. Yeast strains were cultivated at 30 °C with shaking at 220 rpm in Yeast extract Peptone Dextrose (YPD) medium [1% (wt/vol) yeast extract, 2% (wt/vol) peptone, 2% (wt/vol) dextrose, with or without 100 μg mL$^{-1}$ Zeocin] or synthetic complete (SC) medium [0.67% (wt/vol) yeast nitrogen base, 2% (wt/vol) glucose] with appropriate dropout supplements. *A. nidulans* strains were grown at 37 °C or 28 °C with shaking at 220 rpm in CD Medium [1% (wt/vol) glucose, 5% (vol/vol) nitrate salts (12% (wt/vol) NaNO$_3$, 1.04% (wt/vol) KCl, 1.04% (wt/vol) MgSO$_4$•7H$_2$O, 3.04% (wt/vol) KH$_2$PO$_4$), and 0.1% (vol/vol) trace elements (2.2% (wt/vol) ZnSO$_4$•7H$_2$O, 1.1% (wt/vol) H$_3$BO$_3$, 0.5% (wt/vol) MnCl$_2$•4H$_2$O, 0.16% (wt/vol) FeSO$_4$•7H$_2$O, 0.16% (wt/vol) CoCl$_2$•5H$_2$O, 0.16% (wt/vol) CuSO$_4$•5H$_2$O, 0.11% (wt/vol) (NH$_4$)$_6$Mo$_7$O$_{24}$ •4H$_2$O); pH 6.5], or CD-ST Medium [2% (wt/vol) starch, 2% (wt/vol) tryptone, 5% (vol/vol) nitrate salts, and 0.1% (vol/vol) trace elements; pH 6.5] containing appropriate supplements. *Escherichia coli* strain FAST-T1 was used for plasmid construction. *E. coli* strains were grown in LB medium [0.5% (wt/vol) yeast extract, 1% (wt/vol) tryptone, 1% (wt/vol) NaCl] at 37 °C with shaking at 250 rpm, with or without 50 μg mL$^{-1}$ ampicillin.

Plasmid pJET1.2, psgtRNA and pYH-WA-pyrG were used for routine gene knock-in[21,37]. The translation blocking sequence (TBS)[20] was ordered from BGI Genomics (China), and the reporter genes, selection marker genes and homologous arms were amplified from laboratory plasmids, yeasts or *A. nidulans*. These DNA fragments were combined as designed and then ligated with pJET1.2, psgtRNA or pYH-WA-pyrG by In-Fusion cloning technology (Vazyme, China). pRS425 was also ligated with *GFP* and *mCherry* by In-Fusion cloning technology to construct yeast expression vector. The plasmids for GTR-CRISPR-mediated genome editing were assembled with designed gRNA(s) using Golden Gate assembly (New England Biolabs, USA)[21]. All DNA constructs were confirmed by gene sequencing by Sangon (China). All gRNA, plasmids, strains, and primers used in this study are listed in Supplementary Data 3, Supplementary Data 7, Supplementary Data 8, and Supplementary Data 9, respectively. See Supplementary Method 1–3 for additional details.

## Transformation of yeasts and *A. nidulans*

The yeast expression cassettes amplified from corresponding pJET1.2 and psgtRNA-based plasmids or expression vectors were transferred into *S. cerevisiae*, *P. pastoris*, and *Y. lipolytica* using the Frozen-EZ Yeast Transformation II Kit (Zymo Research, USA). For GTR-CRISPR-mediated genomic integration, 0.5 μg pCas-based plasmids containing gRNA(s) and 1 μg kb$^{-1}$ donor DNAs were co-transformed into 100 μL competent cells of *S. cerevisiae* by electroporation[21]. To remove the *ScUra* marker, strains were cultivated overnight in YPD medium and then plated on SC plate containing 1.0 g/L 5-fluoroorotic acid. Colonies were validated by replica plating on YPD and SD-URA plates. For protoplast transformation of *A. nidulans*, 10$^8$ fresh spores were inoculated into 50 mL CD medium and incubated at 30 °C for 12–13 h[38]. The mycelium was collected and washed with Osmotic Medium (1.2 M MgSO$_4$, 10 mM sodium phosphate, pH 6.5), and then inoculated into 10 mL of Osmotic Medium containing 30 mg lysing enzymes (Sigma, USA) and 20 mg Yatalase (Takara, Japan) at 80 rpm for 4–6 h at 37 °C. The cells were overlaid with 10 mL of Trapping Buffer (0.6 M sorbitol, 0.1 M Tris-HCl), and then centrifuged at 4229 × g. The protoplasts were collected from the interface, washed with STC (1.2 M sorbitol, 10 mM CaCl$_2$, 10 mM Tris-HCl), and resuspended in STC. The protoplast suspension was incubated with the transforming DNA for 50 min on ice, followed by addition of 60% PEG 4000 solution and incubation at room temperature for 20 min. The cells were then plated onto solid CD-Sorbitol Medium (CD Medium with 1.2 M sorbitol) and incubated at 37 °C for 2–3 days. The transformants were confirmed by diagnostic PCR.

## Proteomics analysis

*S. cerevisiae* CEN.PK2-1D cells were cultivated in YPD medium at 30 °C with shaking at 250 rpm for 24 h to reach exponential growth phase.

4D label-free qualitative proteomics analysis was performed by PTM BIO (China). After protein extraction, trypsin digestion and LC-MS/MS analysis, the resulting MS/MS data were processed using the MaxQuant search engine (v.1.5.2.8). The resulting databases were searched against all entries of the *S. cerevisiae* proteome (https://www.uniprot.org/taxonomy/4932; downloaded March 2020) concatenated with a reverse decoy database. The mass tolerance for precursor ions was set to 20 ppm in the initial and the main searches, and the mass tolerance for fragment ions was set to 0.04 Da. The false discovery rate was adjusted to <1% and the minimum score for peptides was set to >40. The proteins with intensity-based absolute quantification values >5 × 10$^5$ were selected as potential drivers.

## Preparation of gRNAs and donor DNAs

To amplify donor DNA by one-step PCR, the gRNAs for potential driver genes with CRISPR cleavage sites <5 bp away from the stop codon were predicted by EuPaGDT[39]. To test a library of potential integration sites, donor *GFP* DNA was amplified with primers that also contained ~40 bp of homologous sequences of the endogenous driver target genes, and the *IGG6* and *FLAG* sequences were introduced by the forward primers. *EfmvaE* (GenBank: KX064239.1, https://www.ncbi.nlm.nih.gov/nuccore/KX064239.1/) and *EfmvaS* (GenBank: KX064238.1, https://www.ncbi.nlm.nih.gov/nuccore/KX064238.1) of *Enterococcus faecalis*; *SgSQE1* (encoding a squalene epoxidase), *SgEPH3* (encoding an epoxide hydrolase), *SgCYP87D18* (encoding a cytochrome P450-dependent monooxygenase), and *SgCDS* (encoding a cucurbitadienol synthase) from *Siraitia grosvenorii*; and *AtCPR* (encoding a cytochrome P450 reductase, GenBank: OAO98446.1, https://www.ncbi.nlm.nih.gov/protein/OAO98446.1) from *Arabidopsis thaliana* were synthetized by Sangon (China). *EfmvaE*, *EfmvaS*, *ERG8*, *ERG9*, *ERG12*, *ERG19*, *ERG20* and *IDI1* for the optimized mevalonate pathway; and *AtCPR*, *SgCYP87D18*, *SgCDS*, *SgSQE1*, and *SgEPH3* for the biosynthesis of mogrol from squalene were each amplified from synthetized genes or from *S. cerevisiae* genomic DNA with primers containing ~60 bp homologous fragments of the endogenous driver target genes. The DNA sequence of these 13 genes has been provided in Supplementary Data 10. In addition, the forward primers also included the *IGG6* sequence and the reverse primers also included the peroxisome signal sequence *MDH3* (Supplementary Data 9). Three sets of driver genes showing high, medium and low expression were chosen for *SgCDS*, *SgSQE1*, and *SgEPH3* (Supplementary Table 3). See Supplementary Method 3 for additional details.

## Determination and analysis of protein expression

The parental strains of *S. cerevisiae*, *P. pastoris* and *Y. lipolytica* and their transformants containing reporter genes were cultivated in SC medium with appropriate dropout supplements at 30 °C for 24 h with shaking at 250 rpm. *A. nidulans* strains were cultured in CD-ST medium at 28 °C for 4 days with shaking at 220 rpm. GFP and mCherry fluorescence were observed by an IX73 fluorescence microscope (Olympus, Japan), and fluorescence intensity of GFP was determined by CytoFLEX (Backman Coulter, USA). To assess integration stability, strains with GFP expression cassettes were grown in 800 μL of SC -uracil medium at 30 °C for 24 h. Eight μL of each culture was used to inoculate 800 μL of fresh medium, and this procedure was repeated for ten passages. The fluorescence intensities of the cultures from the first and the last passages were measured using EnSpire (PerkinElmer, USA).

To detect the production of Tdh3p, the *S. cerevisiae* parent strain and its transformants containing *TDH3-FLAG* were cultured in SC medium at 30 °C with shaking at 250 rpm for 24 h. To detect the expression of GFP, *P. pastoris* and its transformants containing *GFP* were cultured in YPD medium for 24 h. Cells in the exponential growth period were homogenized using a FastPrep instrument (MP biomedicals, USA)[40]. The supernatant containing total protein was collected by centrifugation at 12,000 g for 10 min. After sodium dodecyl sulfate-

polyacrylamide gel electrophoresis (SDS-PAGE) with 12% polyacrylamide gels, the proteins were immediately transferred to a polyvinylidene difluoride (PVDF) membrane (BIO-RAD, USA). The anti-FLAG-tag rabbit polyclonal antibody (HX1819, at 1:5000 dilution), the anti-GFP rabbit polyclonal antibody (HX1824, at 1:5000 dilution), the anti-β-tubulin rabbit polyclonal antibody (HX1984, at 1:10,000 dilution), and the anti-GAPDH rabbit polyclonal antibody (HX1832, at 1:10,000 dilution)[41], ordered from Huaxingbio (China), were used to detect the expression of Tdh3p, GFP, β-tubulin and GAPDH, respectively. HRP-goat anti-rabbit IgG(H + L) (Huaxingbio, China, HX2031, at 1:5000 dilution) was used as the second antibody.

To measure the accumulation of the driver-FLAG and GFP proteins, the *S. cerevisiae* parent strain and its transformants containing *driver-FLAG* and/or *GFP* were cultured in SC medium at 30 °C with shaking at 250 rpm for 24 h, and the *GFP-FLAG* transformant was constructed as a control. ELISA kits for FLAG (FineTest, China)[42] and GFP (Cloud-Clone Corp., China)[43] were used to detect FLAG-containing proteins and GFP, respectively.

### Fermentation and extraction of engineered yeasts

Engineered yeasts producing squalene or mogrol were incubated overnight in 3 mL YPD medium at 30 °C with shaking at 250 rpm as seed cultures. One milliliter seed culture was used to inoculate 50 mL of YPD medium with or without 30 μg mL$^{-1}$ terbinafine (Aladdin, China) for producing squalene, or 50 μg mL$^{-1}$ Ro 48-8071 (MedChemExpress, USA) for producing mogrol[44]. After 6 days of cultivation, 10 mL culture of the squalene producer HCS1 strain was collected by centrifugation, and extracted with 1 mL *n*-hexane after shattering the cells by zirconia beads (5 mm) in an MP Fastprep-24 5 G cell disrupter (MP Biomedicals, USA). The sample was separated by centrifugation, and the upper *n*-hexane phase was collected for gas chromatography-mass spectrometry (GC-MS) analysis. The mogrol-producer strains HCM1 to HCM3 were cultivated for 2 days, 2 mL culture was collected by centrifugation, and extracted with 500 μL acetone. The crude extract was evaporated and resuspended in 0.2 mL methanol for analysis using liquid chromatography-high resolution mass spectrometry (LC-HRMS). The tetracistron-containing strain K4 was cultured in SC medium at 30 °C with shaking at 250 rpm for 24 h. Total 2 mL culture was collected by centrifugation, and extracted with 300 μL acetone and 700 μL methanol. After filtration, the samples were analyzed for the production of β-carotene and phytoene by LC-HRMS/MS.

### GC-MS and LC-HRMS analyses

GC-MS analysis for squalene was carried out on a GCMS-QP2010 PLUS mass spectrometer coupled with a gas chromatograph (both from Shimadzu Inc., Japan). One microliter aliquots of the samples were analyzed using a 5MS capillary column (30 m × 0.25 mm, 0.25 μm; Shimadzu Inc., Japan) with helium (1 mL min$^{-1}$) as the carrier gas. The fragmentor voltage was kept at 70 eV (EI), and nitrogen was supplied as the nebulizing and drying gas (280 °C) with a flow rate of 10 L min$^{-1}$, and the pressure of the nebulizer was 10 psi. The data were collected in the full scan mode (*m/z* 50–650).

LC-HRMS analysis for mogrol was carried out on a Waters Xevo G2 series liquid chromatograph coupled with a quadrupole time of flight tandem mass spectrometer using a positive ESI source. One microliter aliquots of the test samples were injected for LC-HRMS analysis (Agilent Zorbax Extend-C18 column, 2.1 × 50.0 mm, 1.8 μm, 100 to 50% CH$_3$CN in H$_2$O with 0.1% formic acid for 13 min, 0.4 mL min$^{-1}$). The scan mode of the mass detector was set as *m/z* 110–1000.

The titers of squalene and mogrol were calculated using calibration curves, which were prepared by monitoring the peak area of serially diluted solutions of commercial squalene (Aladdin, China) and mogrol (MedChemExpress, USA), respectively.

LC-HRMS/MS analysis for β-carotene and phytoene was carried out on an Agilent 1290 Infinity II HPLC equipped with an Agilent QTOF 6530 mass spectrometer operated using a positive APCI source. A total of 20 μL aliquots of the test samples were analyzed using a ACQUITY UPLC BEH C18 column (2.1 × 100 mm, 1.7 μm; Waters, USA) with the mobile phase consisting of solvent A: acetonitrile:methanol (70:30, v/v) and solvent B: H$_2$O 100%[45]. The scan mode of the mass detector was set as *m/z* 100–600 for MS and *m/z* 50–600 for MS/MS. The authentic, commercial β-carotene was from Solarbio (China).

### Growth time course of engineered yeasts

The parental *S. cerevisiae* CEN.PK2-1D and the *TDH3::IGG1-GFP* engineered strain were incubated in 3 mL YPD medium at 30 °C and 250 rpm overnight as seed cultures. The seed cultures were inoculated into 50 mL of YPD medium to reach an initial OD$_{600}$ of 0.05 and grown at 30 °C with shaking at 250 rpm for 3 days. Cell growth was monitored by measuring OD$_{600}$ every 4 h.

### RNA procedures

Yeast cells were grown for 24 h in YPD medium at 30 °C with shaking at 250 rpm to reach the exponential growth phase, then harvested for RNA extraction. Total RNA was isolated using Total RNA Extractor (Sangon, China). Reverse transcription (RT)-PCR was performed using the PrimeScript™ RT kit with gDNA Eraser (Takara, China). RNA-seq was performed by NovoGene (China) on Illumina Novaseq platform.

### Transcriptome analysis

The cleaned sequence reads were aligned to the genome of *S. cerevisiae* S288C (GenBank assembly accession: GCF_000146045.2_R64) using Hisat2 (http://daehwankimlab.github.io/hisat2/), with the reference genome and gene model annotation files downloaded from NCBI. The mapped reads of each sample were assembled by StringTie (v1.3.3b) (http://ccb.jhu.edu/software/stringtie/), read numbers were calculated by Feature Counts v1.5.0-p3, and the Fragments Per Kilobase of transcript sequence per Million base pairs sequenced (FPKM) values were calculated. Differential expression analysis was performed using the DESeq2 R package (1.20.0), and DESeq2 adjusted the *p*-value from wald test using Benjamini and Hochberg method (BH-adjusted *p*-values), which is presented in the column of padj (adjusted *p*-values) in the results object. Genes with padj values of <0.05 and |log2_ratio | >1 were identified as differentially expressed.

### Statistics and reproducibility

Analytical PCRs resolved by agarose gel electrophoresis, Western blots and micrographs produced similar results in three independent replicates. For FI of GFP, each experiment was run with three biological replicates. For ELISA, each experiment was run with two biological replicates. Statistical analysis was performed by one-way ANOVA test with Tukey Pairwise Comparisons using SPSS 22.0 statistical software. For squalene production, each experiment was run with three biological replicates. Statistical analysis was performed by two-tailed t-test using SPSS 22.0 statistical software.

### Reporting summary

Further information on research design is available in the Nature Portfolio Reporting Summary linked to this article.

## Data availability

The raw RNA sequencing data generated in this study have been deposited in the NCBI Sequence Read Archive database under accession number PRJNA821996. Data used for proteomics analysis are available via ProteomeXchange with identifier PXD043249. Source data are provided with this paper.

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

## Acknowledgements

This work was supported by the National Key Research and Development Program of China (2018YFA0901800 to S.S., Y.X., C.L., J.M., Y.Q., and W.S.), the National Natural Science Foundation of China (22277137 and 32070053 to Y.X., 22278024 to S.S.), Tianjin Synthetic Biotechnology Innovation Capacity Improvement Project (TSBICIP-KJGG-009 to S.S. and Q.Y.), the Beijing Advanced Innovation Center for Soft Matter Science and Engineering (to S.S. and Z.L.), VTT Technical Research Centre of Finland (to I.M.), the Agricultural Science and Technology Innovation Program of the Chinese Academy of Agricultural Sciences (CAAS-ZDRW202308 to Y.X. and Q.Y.), and the Central Public-Interest Scientific Institution Basal Research Fund (to Y.X. and L.Z.).

## Author contributions

Q.Yue, J.M., H.Z., Y.X. and S.S. conceived the project. Q.Yue and J.M. designed and performed experiments with guidance from Z.A., Q. Yuan, C.L. H.Z., Y.X. and S.S., and assistance from Y.Q., M.Y., and L.Z.. W.Z., Z.L., and W.S. analyzed the data. Q.Yue, J.M., I.M., Y.X. and S.S. wrote the manuscript with comments from all authors.

## Competing interests

I.M. claims competing interests in TEVA Pharmaceuticals Hungary, which are unrelated to the subject of this study. All other authors declare no competing interest.
