## [Peer Review File · Nature Communications]

A polycistronic system for multiplexed and precalibrated expression of multigene pathways in fungiReviewers' Comments:

Reviewer #1:

Remarks to the Author:

In the manuscript, "HACKing: A novel polycistronic system for the multiplexed, precalibrated expression of multigene pathways in fungi" by Professor Molnar and colleagues, a previously discovered intergenic sequence (IGG) was further optimized and used to develop an approach to express heterologous pathways by hijacking endogenous gene expression via markerless genetic fusions of heterologous genes to native genes. While the intergenic sequence is an interesting genetic element and was shown to work broadly throughout fungi, the paper's main claims go under- to unsubstantiated, as I explain below. The four claims are paraphrased from author's text in the abstract.

Claim 1: Efficient translation of more than one protein from a polycistronic mRNA, by a translational re-initiation.

The first part of the claim is fairly well supported. Related to this point and not addressed by the experimental design is if TDH-GFP fusions are made. The western blot detects the FLAG tag on the C-terminus; controls should show that the lack of other FLAG bands on the western are not artifacts of anti-flag antibody unable to recognize an internal FLAG tag. I was unable to locate the information on the exact antibody used.

The re-initiation mechanism requires some further discussion and experimentation. The interpretation of Figure 2 requires further discussion. The TDH3 western is much weaker for constructs 3 and 4, where TBS is after either the FLAG or IGG. The original assertion that the IGG doesn't affect the TDH3 expression is also called into question by these data. Data later in the paper suggest transcription may be negatively impacted. This hypothesis could explain the observations in Figure 2; it should be raised as a possibility there.

Claim 2: HACKing is a powerful strategy for multigene biosynthetic pathway assemblage in yeast and fungi.

The best idea in here is to use pre-calibrated expression levels for heterologous pathways. This assumes of course one knows what those levels are – perhaps from something like in vitro / cell free expression systems. This is not leveraged in either demonstration of HACKing. Demonstrating the use of bicistrons that leverage the diversity of expression levels will be important to demonstrate the broad impact of this tool.

Perhaps the most attractive feature of the HACKing system is that can be used without the development of refactored genetic parts (Promoters, UAS, etc.). Had the system been applied to a system with fewer or no genetic parts, instead of *S. cerevisiae*, the impact of the work would be much clearer. And if we are being fair about it, a system that lacks such genetic parts often doesn't have a good CRISPR-Cas9 system working, and this method can't be used then.

Claim 3: HACKing was validated by rapidly constructing high performance strains.

Yes, the construction was rapid, but the same strains could be made just about as quickly using more conventional CRISPR-Cas9 mediated integration. Therefore, the speed, while good, is not a defining advantage to the method.

As far as high performing strains, the titers are quite good. The squalene titer is on par with that reported recently in <https://doi.org/10.1186/s13068-022-02208-9>.

Claim 4: HACKing addresses the need for predictability, simplicity, scalability, and speed in fungal

pathway engineering.

The analysis of pathway expression effect on cellular level response compared the HACKed strain to the parent. It is more useful to generate a strain where the heterologous pathway is integrated in neutral sites using similar / identical promoters. The point is not to say how a pathway impacts the cellular response but to show how a pathway integrated in this manner affects the cellular response.

Some minor editorial comments:

Line 175 – yielding should be yielded.

Reviewer #2:

Remarks to the Author:

This manuscript proposed a "HACKing" strategy that enables the precalibrated expression of the target gene in an IGG6-mediated bicistron with CRISPR/Cas9-based genome editing. As a proof of concept, 13 incoming biosynthetic genes were integrated into the endogenous driver sites to produce squalene and mogrol. However, some aspects of developing and applying the strategy have not been addressed or discussed in the manuscript.

1. In the title, the author mentioned the "HACKing" strategy as "a novel polycistronic system". However, most results of this study were just based on the bicistronic expression. In lines 94-98, IGG6-mediated polycistronic cassettes including more than 2 genes were constructed, but only the expression of the last ORF was verified using Zeocin resistance. The expression of the first few genes in the polycistronic cassettes should be also characterized, which may be performed by determining their catalyzed products like lycopene.
2. In the second part of the results "Optimization and characterization of IGG-mediated expression", the author's characterization of the IGG6 sequence is not complete enough, and some characterizations should be supplied. a) Whether there is a proportional relationship between the expression of proteins before and after the IGG6 sequence? b) If the introduction of the bicistronic expression of the second gene will affect the expression of the native gene? In lines 47-50, unchanged growth does not indicate unchanged gene expression. More direct evidence, like two fluorescent proteins verification (10.1039/c4cc08502g), needs to be provided.
3. There is a stronger expression of bicistron mediated by IGG6 sequences relative to IRESs. However, its expression intensity is lower than that of 2A peptides. In addition, 2A peptide can be used for polycistronic expression of more genes and possess a relatively small length (10.1039/c4cc08502g), so it can also be used in the system proposed by the author. Please explain the advantages of the IGG6 sequence.
4. In lines 245-247, the effect of the incoming pathway gene on the endogenous driver gene at the transcription level was analyzed. Is there a similar effect at the translation level? The perturbations of these non-targeted genes certainly will produce an uncontrollable effect on the cell, and how to avoid this.
5. Description about the demonstration of IGG6 in *Aspergillus nidulans*, *Pichia pastoris*, and *Yarrowia lipolytica* is too general (Lines 98-101), and the necessary instructions were not given for Supplementary Fig. 4 and Supplementary Fig. 5.

Other general comments

1. Fig. 3b, the underline of protospacer 5' end miss 1 bp.
2. The full name of HACKing, Highly efficient and Accessible system by CrackKing genes into the genome, should be moved where it first appeared.
3. In lines 200-203, related data was not found.
4. The yellow highlight in Supplementary Table 2 (mentioned in Supplementary Notes) was not found.
5. The header of Supplementary Table 7 has a small error.

Reviewer #3:

Remarks to the Author:

The manuscript presents an interesting new technique (termed HACKing) for the assembly of large gene circuits in fungi. The main results came from the engineering, in the yeast *S. cerevisiae*, of two metabolic pathways to produce squalene and mogrol.

HACKing is based on a short 9-nt-long sequence (IGG6), discovered in the fungus *Glarea lozoyensis*. IGG6 is used together with the CRISPR-Cas9 system to integrate a new gene, in a fungal genome, as the second cistron of a synthetic bicistronic sequence. Its expression is proportional to the promoter upstream of the first gene. Therefore, the method is an interesting alternative to the usage of selective markers or non-coding delta sequences. In general, I agree with the authors when they claim that HACKing will "facilitate sophisticated applications in synthetic biology".

Comments:

- Line 57-58: besides the rate of failures with 2A sequences, it should be also noted that their performance can be increased by placing the "GSG" motif in front of them. According to the paper [Szymczak-Workman, A. L.; Vignali, K. M.; Vignali, D. A. Design and construction of 2A peptide-linked multicistronic vectors. Cold Spring Harbor Protoc. 2012, 2012, 199–204], the ribosome-skipping at these optimized 2As reaches an efficiency close to 100%. Moreover, from Fig. 1A it seems that IGG6 guarantees an efficiency of 50% only in a bicistronic sequence. Perhaps, the comparison between IGG6 and 2A sequences shall be discussed in more detail;
- Figure 1A: the letters "a...d" are not explained in the caption and, probably, are not strictly necessary;
- Figure 3 caption: here you use the term HACKing that, however, is introduced later (in the section starting at line 208). Probably, the figure caption should be revised;
- From Figure 3 again, it is clear that you did not remove the PAM sequence in the donor DNA. Why this choice? If I understood well, both Cas9 and the sgRNAs and expressed on episomal plasmids. How long does it take to remove them from the cells? To produce squalene and mogrol, yeast cells are grown for several days in YPD, which should permit to "clean" the cells of the CRISPR-Cas9 system. What about other kinds of circuits, that demand just 24 to 36 hours to reach the steady state?
- It might be useful to add in the supplementary material the DNA sequences of the enzymes used in the two metabolic pathways engineered in this work.

Reviewer #1

In the manuscript, "HACKing: A novel polycistronic system for the multiplexed, precalibrated expression of multigene pathways in fungi" by Professor Molnar and colleagues, a previously discovered intergenic sequence (IGG) was further optimized and used to develop an approach to express heterologous pathways by hijacking endogenous gene expression via markerless genetic fusions of heterologous genes to native genes.

We would like to thank Reviewer 1 for their overall positive evaluation of our work, and for the very thoughtful and challenging criticisms that helped us strengthen our manuscript.

While the intergenic sequence is an interesting genetic element and was shown to work broadly throughout fungi, the paper's main claims go under- to un- substantiated, as I explain below. The four claims are paraphrased from author's text in the abstract.

Claim 1: Efficient translation of more than one protein from a polycistronic mRNA, by a translational re-initiation. The first part of the claim is fairly well supported.

Thanks for emphasizing that this claim is well supported.

Related to this point and not addressed by the experimental design is if TDH-GFP fusions are made. The western blot detects the FLAG tag on the C-terminus; controls should show that the lack of other FLAG bands on the western are not artifacts of anti-flag antibody unable to recognize an internal FLAG tag. I was unable to locate the information on the exact antibody used.

We have now conducted additional experiments to show that the anti-FLAG antibody is able to detect the tag at the C-terminus and also in the middle of fusion proteins, as recommended by the Reviewer. A Tdh3p-FLAG-GFP fusion protein (with no IGG sequence between the fusion partners) produced a GFP signal, and was detected only as a 64 kDa band by the anti-FLAG antibody (Lines 159-161, Supplementary Fig. 1). We also constructed a *TDH3::GFP* (with a stop codon but no IGG) transformant (Supplementary Table 9, Supplementary Fig. 1), and detected only a 37 kDa band indicating the production of Tdh3p-FLAG (Supplementary Figs. 1a), but no GFP signal was observed (Supplementary Fig. 1b).

We further confirmed these points using an *mCherry-fusion-GFP* transformant without IGG between *mCherry-MDH3* (no stop codon) and *GFP*, where both the signal of mCherry and GFP were observed and colocalized in the cell nucleus. Compared to this, in an *mCherry::GFP* transformant without IGG between *mCherry-MDH3* and *GFP*, but with a stop codon following *mCherry-MDH3*, only mCherry was observed localized in the peroxisome, without a GFP signal being detectable (Lines 122-130, Supplementary Fig. 4).

The information about the antibody is now given in the Methods section (Lines 449-452).

The re-initiation mechanism requires some further discussion and experimentation. The interpretation of Figure 2 requires further discussion. The TDH3 western is much weaker for constructs 3 and 4, where TBS is after either the FLAG or IGG. The original assertion that the IGG doesn't affect the TDH3 expression is also called into question by these data. Data later in the paper suggest transcription may be negatively impacted. This hypothesis could explain the observations in Figure 2; it should be raised as a possibility there.

As recommended, we conducted further experiments which showed a moderate reduction of the driver genes upon “hitching” the transgenes (Lines 210-216, Supplementary Fig. 11b). Instead of comparing strains where the TBS would be at different positions, we elected to directly compare the expression of the driver protein in several driver-FLAG transformants and the same in the corresponding *driver-FLAG::IGG6-GFP* bicistronic transformants, upon quantification by ELISA. The accumulation of the driver proteins in the *driver-FLAG::IGG6-GFP* bicistronic transformants were moderately reduced, reaching 38-64% of that in their corresponding driver-FLAG transformants (Supplementary Fig. 11b).

Claim 2: HACKing is a powerful strategy for multigene biosynthetic pathway assemblage in yeast and fungi. The best idea in here is to use pre-calibrated expression levels for heterologous pathways. This assumes of course one knows what those levels are – perhaps from something like in vitro / cell free expression systems.

Instead of in vitro/cell free systems whose reliability to predict the expression of genes under desired fermentation process conditions in vivo would be questionable, we have used a proteomics experiment to select drivers (Line 192 onwards). We of course do not know (and cannot know in advance) for a heterologous pathway what the ideal expression levels of each transgene should be. But, with HACKing, we can now design pathway versions where different transgenes would be expressed at higher/lower levels, by simply joining the transgenes to different drivers (as we newly demonstrate for different versions of the mogrol pathway). This is much faster prototyping than refactoring transgenes with new promoters of different strengths.

This is not leveraged in either demonstration of HACKing. Demonstrating the use of bicistrons that leverage the diversity of expression levels will be important to demonstrate the broad impact of this tool.

For both examples (squalene and mogrol), the drivers were chosen based on the experimentally determined expression levels of the drivers. In addition, for mogrol, certain transgenes were deliberately expressed at the highest detected levels within our identified set of driver genes. However, to address the point of the Reviewer directly, we have now experimentally verified that driver genes with higher expression would lead to higher expression of the heterologous pathway. Thus, we created two additional mogrol biosynthetic modules using driver genes showing medium and low expression levels to generate strains HCM2, and HCM3, respectively. The production of mogrol in HCM1 to HCM3 was $1.04 \pm 0.02 \text{ mg L}^{-1}$, $0.54 \pm 0.01 \text{ mg L}^{-1}$, and 0 mg L^{-1} , respectively (Lines 268 onwards, Fig. 4d). This verifies that hitching pathway genes to drivers that are expressed at higher levels (as shown by the proteomics experiment, i.e., by predetermined driver expression levels) leads to higher levels of mogrol production.

*Perhaps the most attractive feature of the HACKing system is that can be used without the development of refactored genetic parts (Promoters, UAS, etc.). Had the system been applied to a system with fewer or no genetic parts, instead of *S. cerevisiae*, the impact of the work would be much clearer. And if we are being fair about it, a system that lacks such genetic parts often doesn't have a good CRISPR-Cas9 system working, and this method can't be used then.*

We agree with the Reviewer's evaluation on the importance of our method not requiring refactoring, and indeed we have highlighted this advantage as one of the distinguishing characteristics of our method (see Line 343 onwards). Establishing CRISPR-mediated integration for a less-characterized organism is likely to be more straightforward than characterizing promoters, terminators, neutral docking sites and so on.

HACKing also relies on the proteomic identification of driver genes with appropriate expression patterns. With less characterized hosts, such proteomics may also be less facile, although rapid genome sequencing and transcriptomics can alleviate such a concern. We would however suggest that prototyping our method in *S. cerevisiae* still sufficiently illustrates the utility and advantages of HACKing.

Claim 3: HACKing was validated by rapidly constructing high performance strains. Yes, the construction was rapid, but the same strains could be made just about as quickly using more conventional CRISPR-Cas9 mediated integration.

We agree that the integration step would be just as quick for conventional CRISPR-Cas9-mediated integration, since this step does not depend on whether integration is after a driver, or at a “neutral locus” (docking site). What we argue is that our method does not require refactoring each gene before such an integration step, as the Reviewer already noted above, and thus saves multistep cloning. Nor do we need multiple “neutral” sites to be identified and validated for inserting pathways. In addition, our method should not deplete regulators etc. binding to upstream regulatory sequences in front of important native genes (due to the cloning of these sequences to drive the transgenes).

Therefore, the speed, while good, is not a defining advantage to the method.

We would like to respectfully argue that our method allows the intended complex, multiplexed pathway engineering to be achieved by using donor DNA fragments produced by PCR on synthetic gene templates. The primers necessary to achieve this are within the range that are commercially available, even if they encode various localization tags for the transgenes. Compared to other methods that necessitate the introduction of longer “coupling” sequences (18-22 amino acids, i.e., 54-66 bp for 2A peptides), our method allows integrations without additional cloning steps, thus maximizing the speed of engineering.

As far as high performing strains, the titers are quite good. The squalene titer is on par with that reported recently in <https://doi.org/10.1186/s13068-022-02208-9>.

Thank you for drawing our attention to this reference to support our high production of squalene. We have now cited this paper as Ref. 26.

Claim 4: HACKing addresses the need for predictability, simplicity, scalability, and speed in fungal pathway engineering. The analysis of pathway expression effect on cellular level response compared the HACKed strain to the parent. It is more useful to generate a strain where the heterologous pathway is integrated in neutral sites using similar / identical promoters.

While this would be an interesting comparison, the suggested experiment would mean: 1. factoring 13 genes with 13 different promoters and terminators. 2. arranging these into multigene cassettes (which ones should go together? how many genes should be incorporated into one multigene cassette?). 3. integrating the multigene cassettes to neutral sites (how many neutral sites? which ones?). 4. extensive quantitative, comparative fermentation experiments with enough statistical power to be meaningful. We would like to respectfully argue that this would necessitate an excessive amount of work that could not be completed and presented in the present manuscript. Further, the above-listed variables would mean that we would never have an “ideal” control for the HACKing system, i.e., we would need to engineer multiple such controls with different neutral landing sites etc., making such comparisons unfeasible.

The point is not to say how a pathway impacts the cellular response but to show how a pathway integrated in this manner affects the cellular response.

In addition to the previous argument, we would like to respectfully suggest that such comprehensive metabolomic/physiological comparisons would necessitate perhaps several FTE years to complete, if at all feasible. We have presented data to quantify how the expression of the hitched transgenes affect the expression of the drivers, and the immediate genomic neighbors of the drivers (Lines 308 onwards, Supplementary Fig. 13). These are immediate and local effects that are feasible to correlate to our engineering. Evaluating the general cellular response would not only be more difficult, but it would also be very difficult or even impossible to disentangle IGG6-related effects on drivers; the knock-on effects of the modulated enzyme levels of the drivers; the metabolic changes related to diverting of precursors, reducing equivalents, and energy carriers towards the new engineered pathways; the effects related to the presence of the new metabolites; etc.

Some minor editorial comments:

Line 175 – yielding should be yielded.

As recommended, “yielding” has been corrected to read “yielded” (Line 201).

Reviewer #2

This manuscript proposed a “HACKing” strategy that enables the precalibrated expression of the target gene in an IGG6-mediated bicistron with CRISPR/Cas9-based genome editing. As a proof of concept, 13 incoming biosynthetic genes were integrated into the endogenous driver sites to produce squalene and mogrol. However, some aspects of developing and applying the strategy have not been addressed or discussed in the manuscript.

1. In the title, the author mentioned the “HACKing” strategy as “a novel polycistronic system”. However, most results of this study were just based on the bicistronic expression. In lines 94-98, IGG6-mediated polycistronic cassettes including more than 2 genes were constructed, but only the expression of the last ORF was verified using Zeocin resistance. The expression of the first few genes in the polycistronic cassettes should be also characterized, which may be performed by determining their catalyzed products like lycopene.

We would like to thank Reviewer 2 for their overall positive evaluation of our work, and for the thoughtful suggestions to improve our characterization of the system and that of our presentation.

As recommended, we have now detected the products of the carotenoid pathway in the engineered yeast strain K4 with the tetrascistron containing *crtYB*, *crtE*, *crtI*, and *KanR* (Lines 135-138). We include LC-HRMS/MS data as Supplementary Fig. 6 that validates the production of β -Carotene and phytoene in K4, verifying the expression of the first 3 ORFs, in addition to that of Zeocin resistance (*KanR*, fourth gene in the operon, Supplementary Fig. 5).

2. In the second part of the results “Optimization and characterization of IGG-mediated expression”, the author’s characterization of the IGG6 sequence is not complete enough, and some characterizations should be supplied. a) Whether there is a proportional relationship between the expression of proteins before and after the IGG6 sequence? b) If the introduction of the bicistronic expression of the second gene will affect the expression of the native gene? In lines 47-50, unchanged growth does not indicate unchanged gene expression. More direct evidence, like two fluorescent proteins verification (10.1039/c4cc08502g), needs to be provided.

As recommended, these issues are now addressed in the “Bicistronic expression of genes of interest” section of the results, including large amounts of new data.

As for issue a) (proportion of the expression of the driver and the hitched gene), we show that the expression strengths are correlated between the two partner proteins (Lines 204-206, Supplementary Fig. 10). We also present new data that correlates ELISA-based protein expression levels of selected drivers and their hitched GFP partners (Lines 206-210 and Supplementary Fig. 11a), showing that 75% of the selected *driver-FLAG::IGG6-GFP* transformants express the GFP at about 31-76% of that of the driver.

As for issue b) (effects of hitching on the driver gene), we also supply new data (Lines 211-216, Supplementary Fig. 11b): “To evaluate the effect of *IGG6-GFP* introduction on the expression of the driver protein, several *driver-FLAG* transformants were also constructed using CRISPR/Cas9, and the expression of the driver protein in these transformants and their corresponding *driver-FLAG::IGG6-GFP* bicistronic transformants were quantified by ELISA. The accumulation of the driver proteins in the *driver-FLAG::IGG6-GFP* bicistronic transformants were moderately reduced, reaching 38-64% of that in their corresponding *driver-FLAG* transformants (Supplementary Fig. 11b)”.

3. *There is a stronger expression of bicistron mediated by IGG6 sequences relative to IRESs. However, its expression intensity is lower than that of 2A peptides. In addition, 2A peptide can be used for polycistronic expression of more genes and possess a relatively small length (10.1039/c4cc08502g), so it can also be used in the system proposed by the author. Please explain the advantages of the IGG6 sequence.*

As recommended, the advantages of the *IGG6* sequence vs. 2A peptides have now been discussed in Lines 326-337 in the Discussion section: “Although this expression intensity is lower than that mediated by some 2A peptides, *IGG6* still offers distinct advantages for polycistronic expression of more genes. First, *IGG6*-mediated translation does not append any additional amino acids that could affect the functions of either protein encoded by the bicistron ¹¹. Second, *IGG6* is much shorter than 2A peptides which usually contain 18-22 amino acids ¹². In many cases, various tags, such as those for subcellular localization, need to be appended to the GOI. The DNA sequences encoding 2A peptides are too long to be added to the primers that also encode long tags, such as those for the mitochondrial localization tag MLS26 ^{35,36}. This then necessitates multistep cloning procedures, instead of a single-step PCR amplification of the GOI for CRISPR/Cas9-mediated integration. Third, the viral origin of 2A peptides may raise regulatory concerns, not applicable to *IGG6* that originates from the nonpathogenic fungus *Glarea lozoyensis*.”

4. *In lines 245-247, the effect of the incoming pathway gene on the endogenous driver gene at the transcription level was analyzed. Is there a similar effect at the translation level? The perturbations of these non-targeted genes certainly will produce an uncontrollable effect on the cell, and how to avoid this.*

To evaluate the effect of introduction of bicistronic expression of the second gene on the expression of the native (driver) gene, we conducted new experiments. Several *driver-FLAG* transformants were newly constructed using the CRISPR/Cas9 tool (Supplementary Table 9), and the expression of the driver gene in these transformants and the same in the corresponding *driver-FLAG::IGG6-GFP* transformants were detected and quantified by ELISA using an anti-FLAG antibody. The accumulation of driver proteins in *driver-FLAG-IGG6-GFP* transformants were about 38-64% of that in their corresponding *driver-FLAG* transformants (Supplementary Fig. 11b), suggesting that the expression of driver genes was only moderately reduced by the bicistronic expression of the second gene. These experiments are now described in Lines 210-216. The perturbations of cell growth due to such changes do not seem to be significant, based on the growth curves e.g, in Supplementary Fig. 2.

5. *Description about the demonstration of IGG6 in Aspergillus nidulans, Pichia pastoris, and Yarrowia lipolytica is too general (Lines 98-101), and the necessary instructions were not given for Supplementary Fig. 4 and Supplementary Fig. 5.*

Details on the strains, cultivation conditions, and transformation methods are in the Methods (Line 367 onwards and Line 395 onwards), while the target genes are specified in lines 139-142. Construction of the plasmids is now described in a new section (Line S146 onwards) in the Supplementary Methods section, as recommended.

Other general comments

1. *Fig. 3b, the underline of protospacer 5' end miss 1 bp.*

Fig. 3b has been corrected as recommended, thanks for noticing this mistake.

2. The full name of HACKing, Highly efficient and Accessible system by Cracking genes into the genome, should be moved where it first appeared.

The full name of HACKing has now been added to where it first appears in the Abstract (Lines 35-36) and the main text (Lines 238 and 244), as recommended.

3. In lines 200-203, related data was not found.

In the Source Data file, the "Fig. 3c" sheet has 2 tables: Fig. 3c-1 and Fig. 3c-2. Fig. 3c-2 contains the related data.

4. The yellow highlight in Supplementary Table 2 (mentioned in Supplementary Notes) was not found.

The yellow highlight had been inadvertently deleted, but now it is added back in Supplementary Table 2, as recommended.

5. The header of Supplementary Table 7 has a small error.

The header of Supplementary Table 7 has been corrected, as recommended.

Reviewer #3

*The manuscript presents an interesting new technique (termed HACKing) for the assembly of large gene circuits in fungi. The main results came from the engineering, in the yeast *S. cerevisiae*, of two metabolic pathways to produce squalene and mogrol. HACKing is based on a short 9-nt-long sequence (IGG6), discovered in the fungus *Glarea lozoyensis*. IGG6 is used together with the CRISPR-Cas9 system to integrate a new gene, in a fungal genome, as the second cistron of a synthetic bicistronic sequence. Its expression is proportional to the promoter upstream of the first gene. Therefore, the method is an interesting alternative to the usage of selective markers or non-coding delta sequences. In general, I agree with the authors when they claim that HACKing will “facilitate sophisticated applications in synthetic biology”.*

We would like to thank the Reviewer for their favorable evaluation of our manuscript, and the thoughtful suggestions to improve our presentation.

Comments:

- Line 57-58: besides the rate of failures with 2A sequences, it should be also noted that their performance can be increased by placing the “GSG” motif in front of them. According to the paper [Szymczak-Workman, A. L.; Vignali, K. M.; Vignali, D. A. Design and construction of 2A peptide-linked multicistronic vectors. Cold Spring Harbor Protoc. 2012, 2012, 199–204], the ribosome-skipping at these optimized 2As reaches an efficiency close to 100%. Moreover, from Fig. 1A it seems that IGG6 guarantees an efficiency of 50% only in a bicistronic sequence. Perhaps, the comparison between IGG6 and 2A sequences shall be discussed in more detail;

As recommended, we have rewritten the indicated sentence in the Introduction, and cited the recommended paper (Lines 60-62). Further, we now discuss the comparison between IGG6 and 2A sequences in the first paragraph of Discussion in more detail.

- Figure 1A: the letters “a...d” are not explained in the caption and, probably, are not strictly necessary

In Figure 1a, the letters “a...d” show the statistically significant differences between variables. To clarify this, “Means that do not share a letter are significantly different ($P < 0.05$).” has been added to the legend of Figure 1a.

- Figure 3 caption: here you use the term HACKing that, however, is introduced later (in the section starting at line 208). Probably, the figure caption should be revised;

The Figure 3 caption has been revised to avoid this problem, as recommended.

- From Figure 3 again, it is clear that you did not remove the PAM sequence in the donor DNA. Why this choice?

We have inserted the following explanation to the Supplementary Information (Lines 52 onwards): “For all gene integrations designed as shown in Fig. 3, the PAM region is located close to the stop codon and often within the ORF of the endogenous driver gene. Thus, preserving the sequence of the driver gene means that the PAM sequence is often not eliminated during our engineering. However, the length between the cleavage site and the stop codon was designed to be at most 5 base pairs (Supplementary Table 3), which means that after the donor DNA is integrated, at most 8 base pairs would be left from the protospacer. Considering that a contiguous stretch of at least 13 base pairs is required for annealing

between the gRNA and the target DNA site proximal to the PAM for efficient target cleavage [Jinek, M. et al. A programmable dual-RNA-guided DNA endonuclease in adaptive bacterial immunity. *Science* 337, 816-821 (2012)], the remaining PAM-protospacer sequence is too short to be functional.”

If I understood well, both Cas9 and the sgRNAs are expressed on episomal plasmids. How long does it take to remove them from the cells? To produce squalene and mogrol, yeast cells are grown for several days in YPD, which should permit to “clean” the cells of the CRISPR-Cas9 system. What about other kinds of circuits, that demand just 24 to 36 hours to reach the steady state?

We used 5-FOA to remove episomal plasmids with the *ScUra* marker, as we now describe in lines 400-403. Yes, generally 2-3 days would be enough in YPD to cure the yeast cells, as the Reviewer notes.

- It might be useful to add in the supplementary material the DNA sequences of the enzymes used in the two metabolic pathways engineered in this work.

The DNA sequences have been added to the Supplementary material under Section 4, as recommended.

Reviewers' Comments:

Reviewer #1:

Remarks to the Author:

Thanks for the author's careful and diligent replies. They did a significant amount of additional work and clarified the comparison of their polycistronic system to others. I believe this manuscript is now appropriate for publication.

Reviewer #2:

Remarks to the Author:

The authors have addressed all my critiques in the previous review.

Reviewer #3:

Remarks to the Author:

I am satisfied with the modifications made on the manuscript.

Reviewer #1:

Thanks for the author's careful and diligent replies. They did a significant amount of additional work and clarified the comparison of their polycistronic system to others. I believe this manuscript is now appropriate for publication.

We would like to thank the Reviewer for their constructive feedback and positive evaluation of our work.

Reviewer #2:

The authors have addressed all my critiques in the previous review.

We would like to thank the Reviewer for their helpful feedback and support for the manuscript.

Reviewer #3:

I am satisfied with the modifications made on the manuscript.

We would like to thank the Reviewer for their constructive feedback and favorable evaluation of our work.